# Ultraviolet-Assisted Modified Delignified Wood with High Transparency

**Xiaoli Chen** [†], **Shangjie Ge-Zhang** [†] (iD), **Yu Han, Hong Yang** (iD), **Wenao Ou-Yang, Haotong Zhu, Junyi Hao** and **Jinxin Wang** *(iD)

Department of Physics, College of Science, Northeast Forestry University, Harbin 150040, China; cxl19@nefu.edu.cn (X.C.); gzsj19@nefu.edu.cn (S.G.-Z.); hanyu5214@nefu.edu.cn (Y.H.); yh2020@nefu.edu.cn (H.Y.); oywa19@nefu.edu.cn (W.O.-Y.); 2019212493zht@nefu.edu.cn (H.Z.); hjy2019@nefu.edu.cn (J.H.)
* Correspondence: wangjinxin@nefu.edu.cn
† These authors contributed equally to this work.

**Abstract:** The substrate of solar cells with high haze, transparent, flexible, green and low coatings will be needed in the future. This paper reports a method for ultraviolet-assisted delignification of wood in an alkaline solution environment to improve the transmittance of "transparent wood". Scanning electron microscope (SEM), X-ray diffraction image (XRD), Fourier transform infrared (FTIR) spectroscopy and transmittance-haze and chemical composition analysis were used to explore the mechanisms underlying the effect of ultraviolet-assisted lignin modification on the optical properties of "transparent wood". The results show that UV-assisted delignification accelerates the rate of removal of lignin and chromogenic groups, which in turn improves the optical properties of the "transparent wood", with UV-assisted lignin modification for 2 h increasing the light transmission of the "transparent wood" by 20%. UV-assisted delignification for 4 h and impregnation resulted in "transparent wood" with a transparency of 71% and a haze of 90%. This report provides a rapid and easy method to prepare high-quality "transparent wood". The "transparent wood" with high transmittance and high haze is a potential candidate for transparent solar substrates. Meanwhile, this method is enlightening for high quality, fast and green preparation of other derived functional materials based on lignin wood.

**Keywords:** wood; ultraviolet rays; delignification; transmittance

## 1. Introduction

Replacing petroleum-based materials with bio-based materials is an interesting opportunity as energy demand and environmental protection requirements increase [1–3]. Global warming caused by the massive emission of greenhouse gases such as carbon dioxide has become one of the greatest challenges facing humankind in the 21st century. Carbon neutrality will provide a solution to this challenge of global warming. Therefore, biomass-based composites and solar power are desirable in order to avoid excessive consumption of oil resources and to reduce the carbon footprint [4]. On the one hand, with the growing concern for global ecology, wood-based resource materials—which are widely found in nature, have a large storage capacity, and are renewable and recyclable—are receiving increasing attention [5–7]. On the other hand, solar power has the advantages of being environmentally friendly, safe, inexhaustible and efficient [8], and holds the promise of helping to achieve the environmental goal of carbon neutrality. The development of green materials using renewable resources, combining strength with functional properties, is essential for sustainable development [1]. "Transparent wood" is a new type of material with excellent properties of low thermal conductivity, high modulus, high strength wood, and transparent optical properties. These advantages show great promise in the fields of construction, light-emitting materials, photovoltaic devices, magnetic materials and

energy storage materials. In recent years, many scholars in various countries have been conducting research on the environmentally friendly, rapid and large-scale preparation of highly transparent wood and its application to specific devices [9–11]. "Transparent wood" with low density, high optical transmittance, and low thermal conductivity is a potential candidate for transparent solar cell substrates [12].

The internal refractive index mismatch of wood and the light-absorbing properties of lignin render natural wood opaque [13]. Delignification treatment can effectively remove coloured lignin group from wood and filling the pores with transparent polymers with refractive indices close to those of the delignified wood-based framework can reduce the scattering and refraction of light passing through the interior, so the preparation of "transparent wood" can be divided into two parts: delignification and filling with polymers [14]. In wood delignification, studies have found that lignin absorbs 80–95% of the overall light absorption by wood [15]. To obtain "transparent wood", depending on the oxidising agent used in the delignification process, delignification treatments can be broadly classified into several categories: sulphite, hypochlorite and chlorite [16]. Fink treated wood with a 5% aqueous solution of sodium hypochlorite for 1–2 h to remove coloured substances, including lignin [17]. Berglund's group obtained delignified wood through a modified sodium chlorite method at 80 °C for 6 h. The lignin content was strongly decreased from around 25% to less than 3% [18]. Other researchers have also adapted the $NaClO_2$ method [19,20]. Hu's group removed lignin by cooking in NaOH and Na2SO3 solution, followed by hydrogen peroxide treatment, resulting in a lignin content lower than 3% [21]. Qiu et al. used sodium chlorite to remove lignin from balsa wood chips to prepare flexible and transparent phase change materials with thermally reversible optical properties [22]. These methods are energy intensive and prone to waste of resources, and the excessive removal of lignin causes a rapid decrease in the mechanical properties of the wood [23]. As a green oxidant, hydrogen peroxide can also be used to modify lignin to achieve the purpose of decolourisation [7]. Hu et al. prepared "transparent wood" with high light transmittance (>90%) and high haze (>60%) that could be patterned by applying hydrogen peroxide to the wood surface with the aid of sunlight [24]. However, the effect of sunlight on lignin has not been quantitatively analysed. Removing the chromophore groups of lignin and retaining lignin can maintain the mechanical properties of wood composites and achieve excellent optical properties. In addition, using hydrogen peroxide as the main decolourising substance, Li et al. proposed a new green method which retained 80% of the lignin [25]. The lignin was removed by modifying the lignin to remove the chromophore groups for the purpose of decolourisation. In terms of wood impregnated with polymers, Li et al. successfully impregnated nanoscale cellulose fibre networks on cell walls with refractive index-matched PMMA to obtain optically "transparent wood" with light transmittance of up to 85% and haze of 71% [18]. In addition, Li et al. went deeper and, for the first time, assembled low-temperature (<150 °C) treated perovskite solar cells directly onto "transparent wood" substrates—a practice that resulted in a solar cell power conversion efficiency of 16.8%—thus confirming that "transparent wood" is an ideal substrate for sustainable solar cells [26]. Building on previous work, this paper continues to investigate methods for the removal of lignin from wood to produce highly transparent wood, presenting UV-assisted methods, and attempting to analyse the mechanisms that influence UV-assisted delignification and thus the optical properties of 'transparent wood.

## 2. Materials and Methods

### 2.1. Materials

Samples of Balsa wood were selected from heartwood provided by a plantation in Kunming, Yunnan Province, China. All samples were selected from wood without knots, discolouration or obvious defects and were cut to a size of 20 mm × 20 mm × 1 mm. To avoid the influence of environmental factors, all natural wood samples were stored at 20 °C and 65% relative humidity to achieve hygroscopic equilibrium before the experiment. Anhydrous ethanol (99%) and deionised water were purchased from the Harbin

Junan Medical Glass Wholesale Station. Hydrogen peroxide ($H_2O_2$, 30%), ammonia ($NH_3$, 25%solution), and a UV lamp (385 to 395 nm of wavelength; 30 W power) were used to help remove delignin. The epoxy resin (#128 resin and #polyether amine D230 hardener, Qian Mo Import and Export Ltd, Zhongshan, China) was used to impregnate the wood. All chemical raw materials were of analytical grade and were used without further purification. The material characteristics table in Table 1 clearly shows the above.

**Table 1.** Table of materials properties.

| Materials | Value | Properties |
|---|---|---|
| deionised water | - | - |
| ethanol | 99% | weight ratio |
| $H_2O_2$ | 30% | weight ratio |
| $NH_3$ | 25% | weight ratio |
| resin | #128 | model |
| hardener | D230 | model |
| UV lamp | 30 W | power |
| UV lamp | 385–395 nm | wavelength |

### 2.2. Preparation of Delignified Wood

Several balsa logs cut in the longitudinal direction were rinsed and soaked under ultrasound for 10 min using deionised water as the medium. The delignin solution was prepared in the ratio of hydrogen peroxide:deionised water:ammonia = 5:5:1 and set aside. The cleaned wood samples were then transferred to the delignin solution and left under UV light or in the dark for several hours. This process is the in situ lignin modification process. Once the process is complete, the wood chip samples are first cleaned several times by ultrasonication with deionised water. This is followed by slow washing with a 95% ethanol solution, thus removing residual chemicals and water from the cell pores to obtain the delignified wood (DW) after the washing is complete. During this process, the sample changes from light brown to white or pale yellow. The DW samples are stored in ethanol pending further use. We obtained eight groups of delignified wood. The wood chips were soaked in the delignin solution and left to stand for 2 h, 4 h, 6 h and 8 h under UV light to obtain the delignified wood denoted as DW-A-U, DW-B-U, DW-C-U and DW-D-U, and for 2 h, 4 h, 6 h and 8 h in the dark to obtain the delignified wood denoted as DW-A, DW-B, DW-C and DW -D.

### 2.3. Preparation of the Impregnating Solution

Mix the epoxy resin and the curing agent at a mass ratio of 3:1 and stir with a magnetic stirrer for 10 min to mix the epoxy resin and the curing agent evenly. The impregnating solution is used immediately to prevent solidification and, thus, difficulties filling in delignified wood cells. The 8 groups of delignified wood (DW) are then placed in a number of plastic round caps and poured with the appropriate amount of the prepared impregnating solution. The DW is placed separately to avoid interference with each other during the vacuum impregnation process. Finally, the DW is pressed against the bottom of the round lid using the two corners of a stainless-steel clamp, which prevents the wood chips from floating on the surface of the impregnating and causing poor impregnation results.

### 2.4. Vacuum Impregnation

The round caps containing the DW are placed in a vacuum chamber, and the impregnating solution is infiltrated into the DW using a vacuum pump. The amount of vacuum pressure applied and the time of vacuum penetration have an important influence on the penetration of the epoxy resin into the DW. The delignified wood template is placed in a vacuum vessel and is completely vacuum permeated with a pre-polymerised solution under a vacuum pressure of −0.08 to −0.09 Mpa. An optimum penetration time of 3 h is chosen, with pressure release every 1 h to completely remove the ethanol and air from the

wood interior. During this phase, the epoxy resin will slowly soak into the DW's interstices under the applied pressure. After filling, the wood is removed, placed on a smooth tin sheet, and cured at room temperature for 24 h. Four groups of DW-A-U, DW-B-U, DW-C-U and DW-D-U are impregnated and cured separately to give TW-A-U, TW-B-U, TW-C-U and TW-D-U "transparent wood" (TW). DW-A, DW-B, DW-C and DW-D groups are also impregnated and cured separately and named TW-A, TW-B, TW-C and TW-D. In addition, the 3 h vacuum impregnation time of the DW-D-U group was changed to 1 h and 2 h to obtain the TW-D-U1 and TW-D-U2 groups. Detailed flow chart for the preparation of transparent wood as shown in Figure 1.

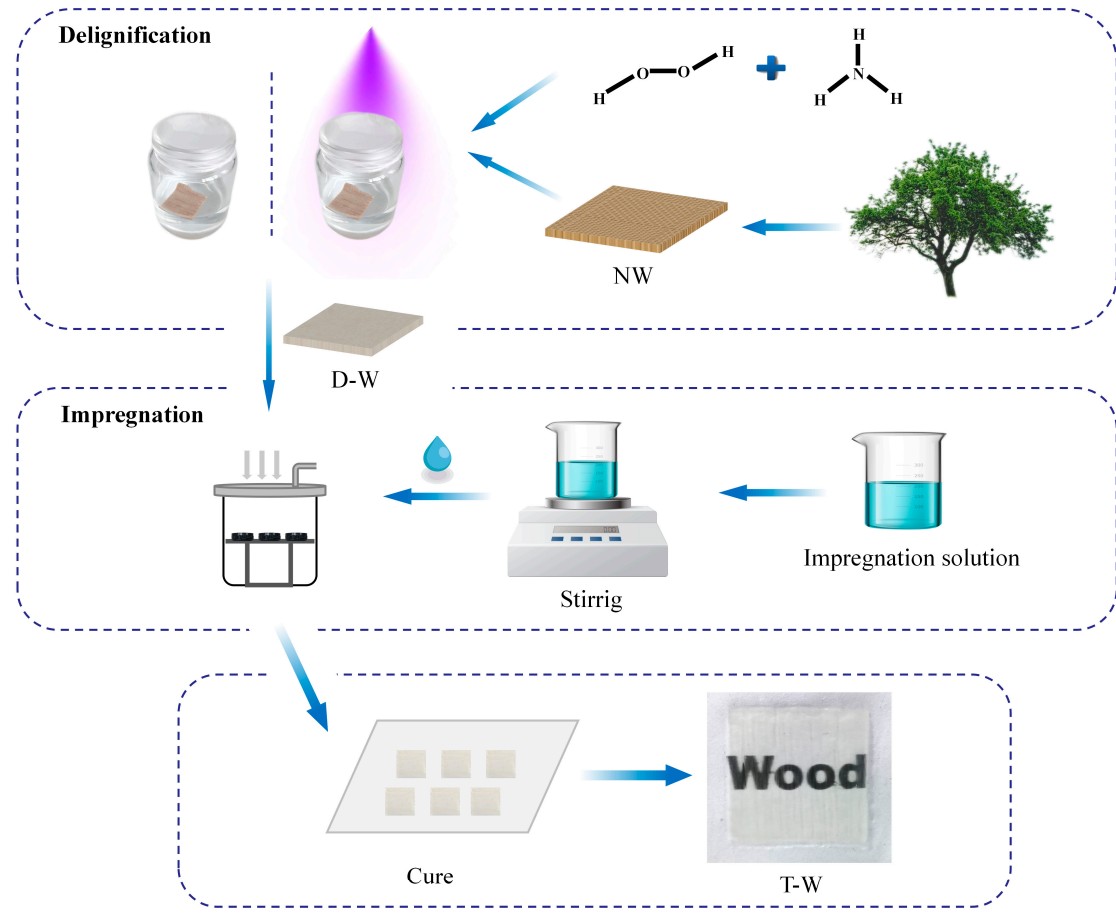

**Figure 1.** A detailed flow chart for the preparation of "transparent wood".

### 2.5. Characterisation

The cross-section SEM images of the samples were observed with a field emission scanning electron microscope (SEM, Hitachi S-4U8020, Hitachi Corporation, Tokyo, Japan) in the secondary electron mode. To obtain SEM micrographs of the cross-section, the surface of the wood piece was lightly scratched with a knife and then broken open along the trace. Measurement of the crystal structure of DW and TW was taken using an X-ray diffractometer (XRD, D8 VDVANCE, Bruker Corporation, Germany). FTIR spectra of DW and TW were recorded using a Nicolet IS10 spectrometer in the range of 400 to 4000 cm$^{-1}$. Transmittance and haze of "transparent wood" were measured by Transmittance-Haze Meter Type HT03 (repeatability < 0.05 for haze and < 0.1 for transmittance, according to ASTMD1003, Coretech Instruments Ltd, Shenzhen, China). We wiped the surface of the transparent wood with alcohol before testing and after the alcohol evaporated, aligned the centre of the transparent wood with the through the light port on the test bench. National

Renewable Energy Laboratory (NREL) analytical methods were used for lignin, cellulose and hemicellulose content testing.

## 3. Results and Discussion

### 3.1. Microstructural Analysis

As shown in Figure 2a,b—which contain scanned electron microscopy images of untreated balsa wood (NW)— wood has a cellular architecture resembling a honeycomb structure at the micrometre scale. The average fibre diameter is 800 microns. They show that the cell cavities are in a hollow state with large filling spaces, and the layered porous structure provides a framework and anchor points for polymer impregnation attachment. After treatment with hydrogen peroxide, the porous structure of the wood was well preserved, but the cell walls were looser than in untreated wood. As shown in Figure 2c,d, there are obvious pores in the cell walls because lignin and pectin were partially removed from the wood after treatment with a hydrogen peroxide solution. Normally, the removal of lignin results in the formation of micropores and nanopores in the cell wall. The resulting micropores facilitate the penetration of the epoxy resin into the wood. Figure 2e,f show the morphology of low and high magnification "transparent wood" (TW). Figure 2e shows that the wood cell crevices are filled with a large amount of polymer, and the polymer is tightly adhered to the wood, the treated wood surface becomes smoother, and some of the original micro-grooves and furrow structures are covered. On closer inspection of Figure 2f, the polymer deposited on the wood can be easily observed. The surface of the sample is covered entirely with polymers, which exhibit deposited aggregates on the wood surface and form a micro-raised structure with a large surface roughness [27]. Some of the cell cavities are well filled with epoxy resin. In contrast, others have small pores within them, which can lead to refraction of light by "transparent wood", reflected macroscopically by high haze and low transmittance.

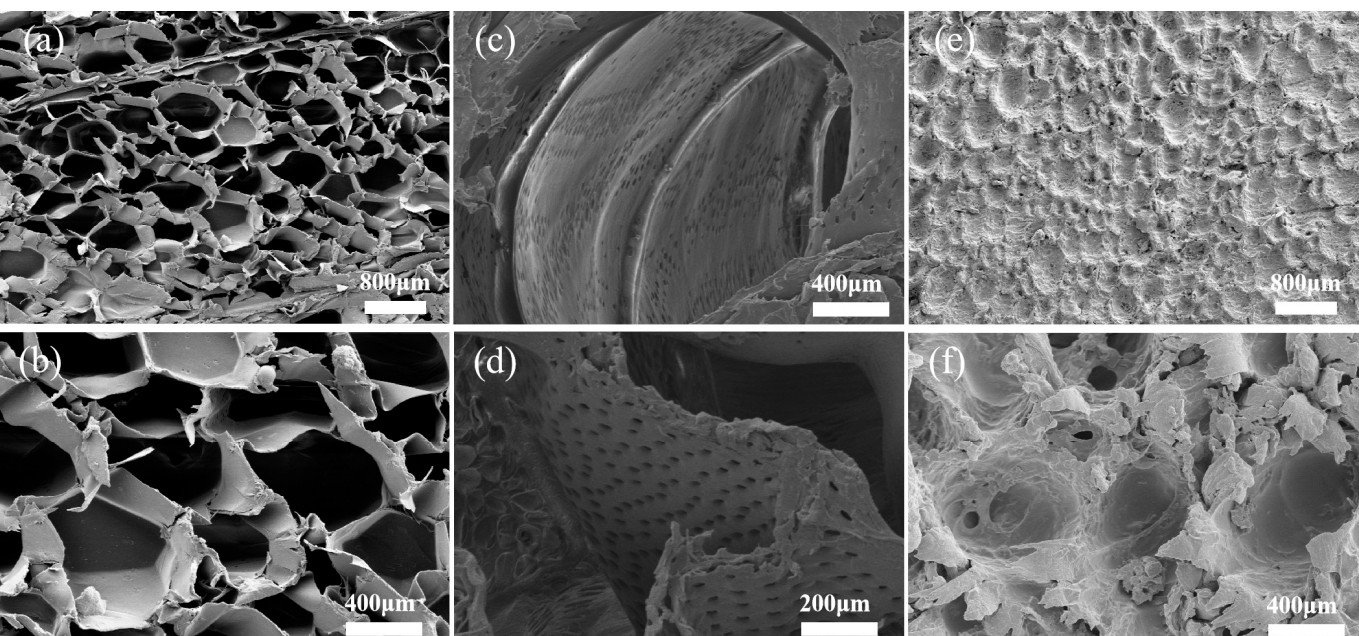

**Figure 2.** SEM images of natural wood (NW)at (**a**) low magnification, (**b**) high magnification, SEM images of "transparent wood" (DW-D-U) at (**c**) low magnification, (**d**) high magnification, SEM images of "transparent wood" (TW-D-U) at (**e**) low magnification, (**f**) high magnification.

Apart from the differences in delignification and impregnation methods, the type of wood chosen and the growing environment play a key role in the treatment results, and the wood dimensions vary with temperature and humidity [28]. The thickness of natural wood

is 1.013 ± 0.064 mm, and that of transparent wood is 1.461 ± 0.155 mm. This is because part of the impregnation solution forms a film on the surface of the wood, and therefore there is a significant increase in thickness.

### 3.2. Analysis of XRD Image

X-ray powder diffractometer (D8 VDVANCE, Germany): Cu Ka-rays (λ = 1.541 80 Å) as diffraction source; operating voltage 40 kV, operating current 40 mA, scanning angle (2θ) range 5° to 90°, step 0.02°. The alteration of the crystal structure was studied while scanning in "reflection" mode.

Cellulose is a macromolecular compound with homogeneous polycrystalline, which consists of a large number of microcrystalline of about 10 nm in an entirely irregular molecular chain [29]. The percentage of cellulose crystal region in the entire cellulose is called the crystallinity of cellulose. It can be seen from Figure 3 that in the XRD diagram of natural wood, the highest peak is located at 2θ = 22.3°. This peak is the 002 crystal plane, which is higher than the peak value of the 101 crystal plane, mainly because the molecules in the cellulose are on the plane of the 002 crystal. The structure of cellulose in wood is complicated. Compared to the natural sample, the 002 crystalline surface peak of DW-D-U is reduced, and the half-height width of the diffraction peak becomes wider. The size of the microcrystals was calculated using the Scherrer equation based on the half-width of the diffraction peak. There is a slight reduction in the grain size of the treated delignified samples compared to the grain size of the original samples. A minor disruption of the structure of the cellulose macromolecules of delignified wood occurred [30,31]. The diffraction images of "transparent wood" indicate that the area where the crystal peak is located is between 17° and 22°, and there is a significant difference between natural wood, indicating that the penetration of epoxy resin changes the internal crystal structure of the wood.

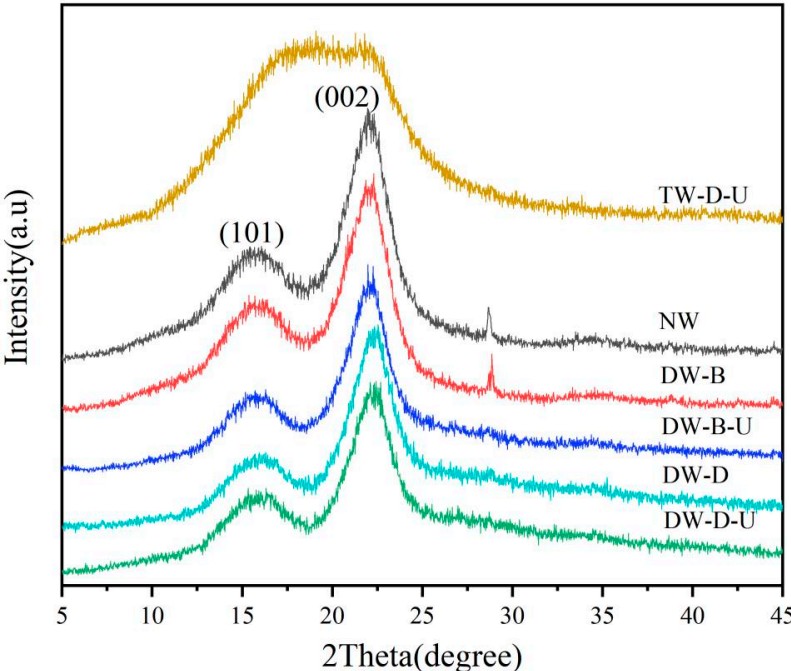

**Figure 3.** XRD images of natural wood (NW), four groups of delignified wood DW-B, DW-B-U, DW-D, DW-D-U and "transparent wood" TW-D-U.

### 3.3. Analysis of Transmittance and Haze

The mean values of transmittance and haze for the four "transparent wood" groups TW-A-U, TW-B-U, TW-C-U and TW-D-U were 55% and 96%, 71% and 90%, 78% and 83%, and 82% and 75%, respectively. At the same time, the transmittance and haze of the four

"transparent wood" groups TW-A, TW-B, TW-C and TW-D were measured to be 34% and 99%, 59% and 94%, 72% and 87%, and 78% and 78%, respectively. As shown in Figure 4a,b, the transmittance of the "transparent wood" increased, and the haze decreased as the lignin modification time increased, both with and without UV assistance. The TW-D-U group achieved the highest value of 82% for transmittance and the lowest value of 75% for haze, having the best optical measurements. In addition, the shorter the delignification time, the more significant the difference in transmittance between "transparent wood" made under UV-assisted and dark conditions; with the "transparent wood" TW-A-U obtained by UV-assisted lignin modification for two hours being 20% more transparent than TW-A, and TW-B-U 12% more transparent than TW-B. It is clear from Figure 4c that DW-B is light yellow, and TW-B is more turbid. Part of the incident light propagation direction changes significantly after passing through TW-B, and the light-absorbing substance absorbs the light, and the 'TW-B' characters become blurred. DW-B-U is whiter than DW-B, while TW-B-U is more transparent and clearer than TW-B. This is due to the accelerated removal of chromophores from the wood lignin by UV irradiation. The increased removal of lignin and the reduction in colour-forming clusters reduces the absorption of light by the wood. As a result, UV-assisted delignified wood can appear white in a shorter period of time, resulting in clearer wood.

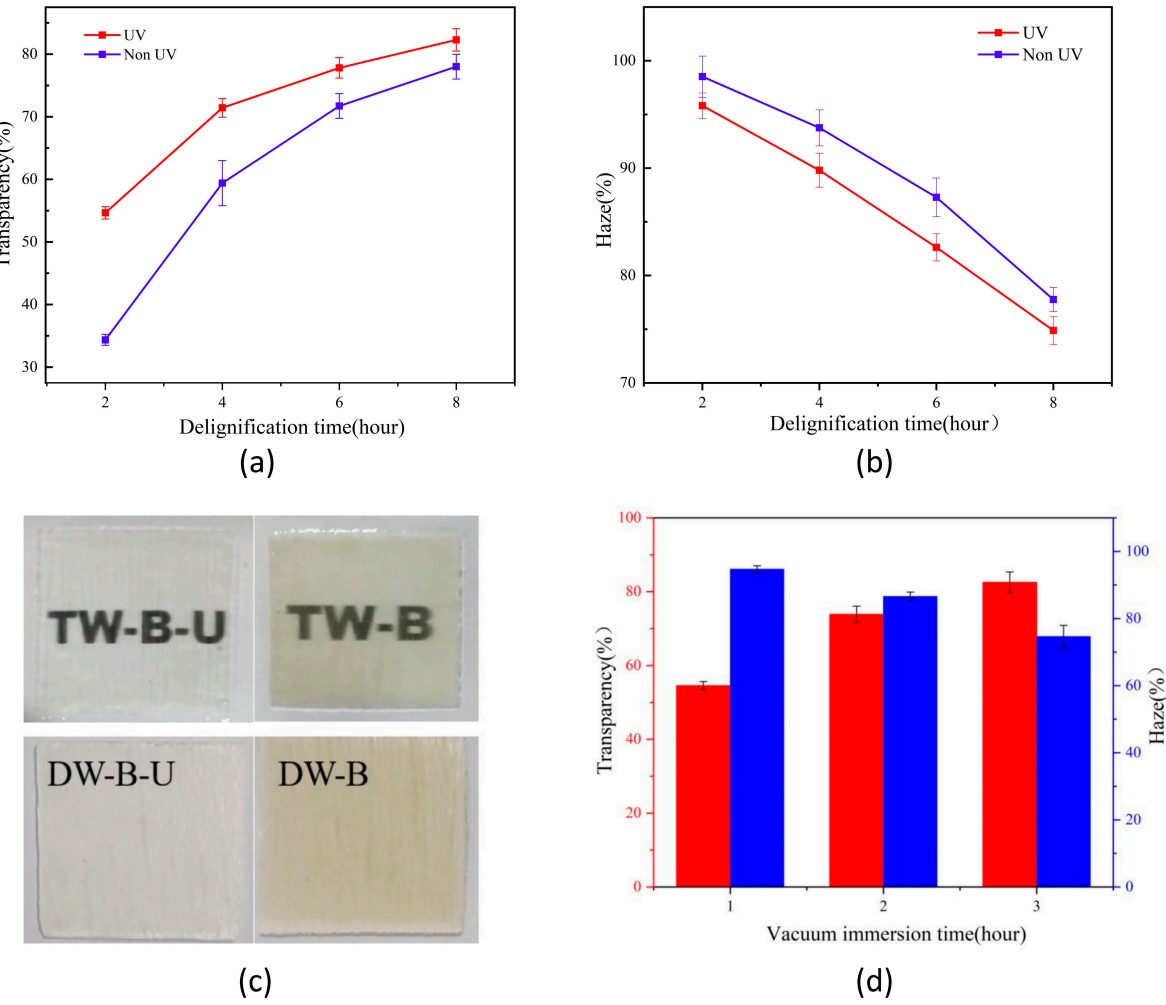

**Figure 4.** Transmittance (**a**) and haze (**b**) of eight groups of "transparent wood" was obtained by soaking the wood in a lignin-modifying solution under UV-assisted or dark conditions for 2 h, 4 h, 6 h and 8 h, followed by a 3 h impregnation treatment, respectively. (**c**) Photographs of TW-B-U, TW-U, DW-B-U and DW-B under daylight source. (**d**) The transmittance and haze of TW-B-U1, TW-B-U2 and TW-B-U were obtained by changing the impregnation time.

In addition, the impregnation time of the "transparent wood" in the TW-D-U group was changed to 1 h and 2 h to obtain the TW-D-U1 and TW-D-U2 groups, respectively. The transmittance and haze were measured to be 55% and 95% for the TW-D-U1 group and 74% and 87% for TW-D-U2. As shown in Figure 4d, the transparency increases with the time of impregnation, and the haze gradually decreases, resulting in a clearer picture through the transparent wood sheet and better optical properties. As the impregnation time increases, the more epoxy resin immerses into the cell and binds closely to the cell wall, the fewer pores become. The degree of refractive index matching inside the transparent material increases; thus, light produces less refraction inside, leading to a decrease in haze and an increase in transmittance [11].

### 3.4. Fourier Transform Infrared Spectroscopy Analysis

The infrared spectroscopic images of balsa wood treated under different conditions are shown in Figure 5. Lignin has a variety of complex structures, and the quinone structure of lignin is the main factor that gives lignin its colour [32]. As seen in Figure 5, compared with natural wood, infrared absorption 1733 cm$^{-1}$ peak of C=O in the quinone structure of lignin in the four groups of delignified wood essentially disappeared, indicating degradation of chromophoric groups [33]. This is due to a reaction between hydrogen peroxide and the quinone structure of lignin, destroying its structure and changing into other colourless structures. The chromophore groups of lignin are removed to whiten the wood [34]. In addition, by comparing the infrared spectra of DW-B-U and DW-B, we found that the spectral lines of the two groups overlap. Still, the peak value of DW-B-U at 1733 cm$^{-1}$ is lower than that of DW-B group, which can indicate that UV irradiation accelerates the removal of chromophore groups in wood to a certain extent. This is also confirmed by the fact that the transmittance of TW-B-U is better than that of TW-B group in the transmittance and haze test mentioned above. In addition, the absorption peak of the aromatic ring skeleton group of lignin decreased slightly at 1504 cm$^{-1}$, the reduced absorption peak at 1231 cm$^{-1}$ of lignin acetyl and ether, and reduced intensity of the peaks at 1031, 1231, 1423 and 1594 cm$^{-1}$. This means that while the chromophore group is removed, a small part of the lignin in the wood is dissolved in the alkaline solution. Compared with the spectra of original wood, the intensity of the infrared spectra of DW-D-U and DW-D samples decreased significantly at 3 332 cm$^{-1}$, indicating that the cellulose structure of lignin delignified wood was damaged to a certain extent [15].

Chemical changes in wood components such as lignin and hydrogen peroxide under alkaline conditions lead to partial hemicellulose and lignin removal, weakening the links between wood tissues. Therefore, although the transmittance of the "transparent wood" is expected to be higher as the lignin modification time continues to increase, the mechanical properties will significantly decrease, which limits the manufacture of large structures [35]. Therefore, this study does not pursue too much higher transmittance.

It can be seen from the infrared spectroscopic images of TW-D-U that the benzene-disubstituted absorption peaks of the benzene-skeleton near 830 cm$^{-1}$ and 1606 cm$^{-1}$ appear in the epoxy resin-filled wood of balsa wood and the vibration absorption peaks of the benzene-skeleton near 1508 cm$^{-1}$ and 1606 cm$^{-1}$. There is an epoxy-based symmetric stretching vibration absorption peak near 1243 cm$^{-1}$ [36]. The results showed that the vacuum-impregnated delignified wood exhibits mainly the characteristic peaks of the epoxy resin without forming new characteristic peaks, indicating that no chemical bond was formed between the wood and epoxy resin. However, a hydrogen bond was formed between the hydroxyl group and epoxy group in the wood.

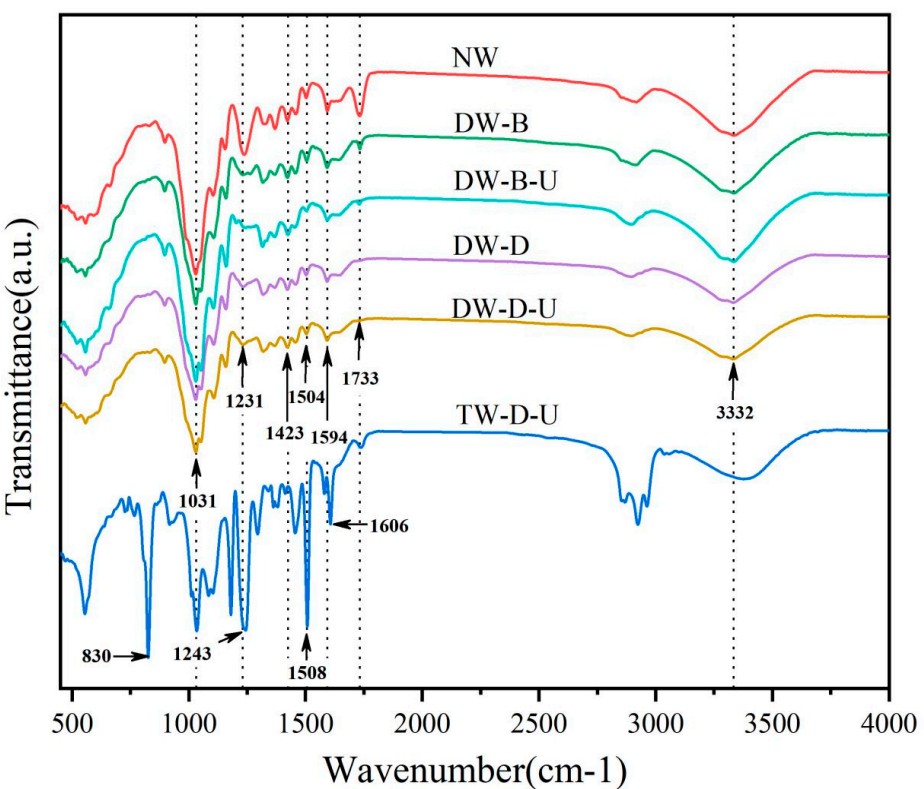

**Figure 5.** Infrared spectral images of natural wood NW, four groups of delignified wood DW−B, DW−B−U, DW−D, DW−D−U and "transparent wood" TW−D−U.

*3.5. Analysis of Lignin Content*

The lignin, cellulose and hemicellulose mass fractions were used as assessment indicators to measure the lignin loss in delignified wood prepared under different treatment conditions in NW, DW-B, DW-B-U, DW-D and DW-D-U. The test results detailed in Table 2 show that in comparing the DW-B-U and DW-B groups or comparing the DW-D-U and DW-D groups, UV irradiation for a certain period of time accelerated the rate of lignin removal, resulting in clear wood with higher transparency. UV-assisted delignification of DW-B-U resulted in 17% higher delignification than the DW-B group, all other things being equal. The lignin content of the treated wood is reduced, while the cellulose content is slightly increased, as the major loss of lignin leads to a relative increase in cellulose content. On the other hand, although there was a significant loss of lignin, up to 45% of the lignin in the DW-B-U group was preserved while being rich in cellulose. This contributes to the mechanical properties of TW-B-U, and the picture in Figure 4c shows that TW-B-U has a high degree of transparency.

**Table 2.** Lignin content, cellulose and hemicellulose content of natural wood NW, four groups of delignified wood DW-B, DW-B-U, DW-D, DW-D-U.

| Groups | LC (wt %) | CC (wt %) | HC (wt %) |
|--------|-----------|-----------|-----------|
| NW | 23.30 | 42.36 | 20.15 |
| DW-B | 17.63 | 43.50 | 18.69 |
| DW-B-U | 14.58 | 44.01 | 17.36 |
| DW-D | 7.78 | 45.22 | 17.94 |
| DW-D-U | 7.26 | 45.19 | 18.12 |

Abbreviations: LC—lignin content, CC—cellulose content, HC—hemicellulose content.

## 4. Conclusions

In this paper, the effect of UV- assisted delignification on the synthesis of optically "transparent wood" with balsa wood as a precursor was measured and analysed. Through the analysis of transmittance and haze results, it was found that under the same treatment time, the transmittance of "transparent wood" prepared by UV-assisted delignification was greatly improved. SEM and XRD results showed that the polymer penetrated the wood and filled the cell cavity under vacuum impregnation treatment. FTIR measurements showed that the chromophores in wood were removed from wood after oxidation by hydrogen peroxide, and UV irradiation greatly increased the rate of this chemical change. The results of transmittance-haze and chemical composition analysis show that the ultraviolet-assisted delignification accelerates the rate of lignin and chromogenic removal, enhancing the optical properties of the "transparent wood". UV-assisted delignification of DW-B-U resulted in 17% higher delignification than the DW-B group, all other things being equal, and the UV-assisted delignification for 2 h increases the transmittance of the "transparent wood" by 20%. After UV-assisted delignification for 4 h up to 45% of lignin is preserved, and then the wood is impregnated to obtain a clear wood with a transparency of 71% and a haze of 90%. At the same time, the "transparent wood" with high transmittance of 82% and haze of 75% was obtained after 8 h of UV-assisted modification of lignin and 3 h of vacuum impregnation-filled polymer. At the delignification stage of clear wood preparation, this paper firstly combines delignification under alkaline solution conditions with UV irradiation and provides a qualitative analysis of the effects produced by UV light. The results of this study show that such transparent materials with light transmittance and high haze can be rapidly manufactured with the help of sunlight with strong UV rays. The universality of the method is attractive for developing sustainable materials for potential solar cell substrate applications. At the same time, this method is enlightening for the high-quality, fast and green preparation of other derived functional materials based on delignified wood.

**Author Contributions:** Conceptualization, X.C. and J.W.; methodology, X.C. and Y.H.; software, X.C.; validation, X.C. and S.G.-Z.; formal analysis, H.Y. and X.C.; investigation, X.C.; resources, X.C.; data curation, X.C.; writing—original draft preparation, X.C. and J.W.; writing—review and editing, X.C. and S.G.-Z.; visualization, X.C. and Y.H.; supervision, J.W. and J.H.; project administration, H.Z. and W.O.-Y. All authors have read and agreed to the published version of the manuscript.

**Funding:** This research was funded by the Fundamental Research Funds for the Central Universities, grant number 2572020BC02.

**Data Availability Statement:** Not applicable.

**Conflicts of Interest:** The authors declare no conflict of interest.

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
