# Peer review of "Ultraviolet-Assisted Modified Delignified Wood with High Transparency"

_applsci, doi:10.3390/app12157406_

Round 1

Author Response

Thank you for your comment. Our response to your comment is attached in the PDF.

Reviewer 2 Report

Review Report 

 Title: Ultraviolet-Assisted Modified Delignified Wood with High Transparency

In this paper signifies method for Ultraviolet-assisted in situ lignin modification of wood in an alkaline solution environment to improve the transmittance of transparent wood and the this report provides a rapid and easy method to prepare high-quality transparent wood.  Authors beautifully investigates the effects of UV irradiation, lignin modification time and impregnation time on the light transmission and haze of transparent wood. And their methodology is clear. The paper is well prepared. However, the article can be further refined by including the following recommendations:

1.     It is suggested to include a table of Materials properties in the Section 2.1 Materials.

2.     The Figure 4 (c & d) needs more explanation and describe the results more clearly.

3.     Rewrite the conclusion to emphasize the work's novelty.

4.     Include some up-to-date related references.

5.     Rewrite the conclusion to emphasize the work's novelty.

Author Response

(The authors gave the same response as above.)

Reviewer 3 Report

In the manuscript of Xiaoli Chen et al. the issue of wood modification in order to obtain materials for ransparent solar substrates is considered. Emphasis is placed on the modification of lignin. The original and modified materials are characterized using SEM, XRD, FTIR methods. I would like to immediately note that it is better to replace the term "transparent wood" used in the article with another one or use quotation marks so as not to mislead the reader. In the literature review, the authors superficially consider the properties of wood and methods for extracting (modifying) lignin. I recommend that this part of the manuscript be expanded to include the benefits of using wood. It is important to note here that wood is a combustible material that changes its geometry with changes in humidity, the chemical composition of wood strongly depends on its type and place of growth, etc. All these factors must be taken into account in It is desirable to expand the list of keywords. Line 70. "xefficiency" - ?! Materials."At a room pressure of -0.8 - 0.9 bar" - this statement needs to be corrected. Figure 1. Needs to be fixed. 2.5 Characterisation. The description of the methods needs to be expanded. Figure 2. It is not clear from the photomicrographs whether the polymer is compatible with the wood. The question also arises about the thickness of the walls, does it increase significantly after processing? 3.2 Analysis of XRD image. This section needs to be redone. The authors should describe in more detail both the method itself and the results obtained, for example, as in the work - https://doi.org/10.3390/fib10050045, identifying the cellulose polymorph, etc. Figure 3. Indexes need to be checked. 3.3 Analysis of transmittance and haze. I recommend rounding up the values.

Author Response

(The authors gave the same response as above.)

Reviewer 4 Report

This paper investigates the effects of UV irradiation, lignin modification time and impregnation time on the light transmission and haze of transparent wood. The statistical analysis,  the results and discussions are well described.

I have some recommendations:

-          In Abstract section, first, must be specified the following terms: SEM, XRD, FTIR

-          In the Introduction section, the purpose of the paper is vaguely presented.

-          Line 78: “… size of 20mm*20mm*1mm…”. It is not an international standard!

-          Line 214: “FTIR “ ??

Author Response

(The authors gave the same response as above.)

Round 2

Reviewer 1 Report

All suggestions were accepted and resolved.

Thanks to the authors.

Reviewer 4 Report

I have noticed that the manuscript has been sufficiently improved to be published in Journal.